# Incidence of Delayed Infections after Lower Third Molar Extraction

**DOI:** 10.3390/ijerph19074028

**Published:** 2022-03-29

**Authors:** Giuseppe Monaco, Maria Rosaria A. Gatto, Gian Andrea Pelliccioni

**Affiliations:** Department of Biomedical and Neuromotor Science, Section of Clinical Dentistry, Alma Mater Studiorum, Università di Bologna, Via San Vitale, 59, 40125 Bologna, Italy; info@ambulatorioarno.it (G.M.); gian.pelliccioni@unibo.it (G.A.P.)

**Keywords:** delayed infection, lower third molar extraction, retrospective study

## Abstract

Purpose: This retrospective study aimed to verify that the onset of delayed infection after lower third molar extraction was influenced by the amount of distal space. Patients and Methods: We evaluated 265 patients (age range 12–55 years), who had one or two mandibular third molars to be extracted. All 380 third molars were removed for orthodontic reasons, periodontal disease, or pericoronitis and were evaluated by the Pell and Gregory classification using the panoramic radiographs. Results: Delayed infection, characterized from purulent exudates from the alveolus and swelling, was reported in 21 extractions between 2 and 8 weeks after surgery. In 16 of the 21 cases of infection, a class III of Pell and Gregory was observed, and this anatomic condition evidenced an extremely reduced space distal to the second molar. Conclusion: This study confirmed that the absence of distal space was significantly correlated with delayed infection. These data are important for proper examination of a patient because, in the case of class I or II of Pell and Gregory, a delayed infection was less likely to occur, while a class III of Pell and Gregory could indicate a greater likelihood of this type of infection.

## 1. Introduction

Third molar extraction can cause immediate (first postoperative week) [1,2,3,4,5,6,7,8,9,10,11,12,13,14,15,16,17] or delayed (from 2 to 8 weeks postoperatively) infections [18,19,20,21,22]. Immediate complications have been extensively studied, and all the factors linkable to the patients [1,2,3,4], tooth impaction [5,6], drug intake, and surgeon technique or experience [7,8,9,10,11,12,13,14,15,16,17] have been well investigated due to the easy correlation with the surgical procedure and the relevance of the prediction of this information for a medicolegal issue. Delayed infection (2–8 weeks from surgery) was more difficult to link to the surgical procedure and, probably for this reason, has been the object of minor attention from researchers [18,19,20,21,22]. In terms of a medicolegal issue, the onset of delayed infection could be a greater problem, because this complication arises in a period in which a patient believes the healing is completed and could disappoint him/her more [22].

The study of Monaco et al. of 2017 [22] showed, in a sample of young patients, this complication was related to a reduced space distally to the second molar, and the percentage of incidence was higher than what had been reported from previous researchers [19,20,21].

The purpose of this retrospective study was to evaluate the importance of the reduced space distal to the second molar in the incidence of delayed infection after removal of an impacted third molar in a large sample of patients with a wide age range.

## 2. Materials and Methods

Ethical approval was obtained from the Ethics Committee of ASL of Bologna (CE 13068).

We evaluated the records of 319 patients for 467 third molar extractions performed by the same surgeon between 2006 and 2016 in Bologna, Italy.

Of this initial sample, only the 265 patients (age range 12–55) who came to the follow-up visits in the first four months post-extraction were considered for the retrospective analysis. These patients (124 male and 141 female) who presented one or two mandibular third molars with a different degree of root formation permitted our team to evaluate 380 extractions.

The extractions were performed for orthodontic reasons or for the presence of at least one of the criteria defined by the National Institute of Health-U.S.A. (N.I.H.) [23] that represent the rationale for third molar removal (pericoronitis, periodontal disease, caries, root resorption, and/or pericoronal cyst).

Third molars extracted for orthodontic reasons (lack of distal space/need for molar distalization) presented, at the radiographic evaluation, incompletely formed roots. These extractions (germectomies) were 218 in 134 patients (68 male and 66 female).

Patients were nonsmokers or declared to smoke less than 10 cigarettes a day.

Third molars extracted for the presence of at least one of the criteria defined from the N.I.H. presented, at the radiographic evaluation, completely formed roots that in 25 cases had a close relationship between the third molar root and the mandibular canal as documented by a TC-Cone-Beam. There were 131 patients with completely formed third molar roots (56 male and 75 female) with 162 extractions. These teeth were partially (102) or completely (60) impacted.

All the teeth evaluated in this study were analyzed according to the Pell and Gregory classification [24] (Figure 1).

### 2.1. Preoperative Care

All the extractions were performed with antibiotic prophylaxis [4] (2 g of amoxicillin and clavulanic acid 1 h before surgery). In patients allergic to penicillin, we prescribed azithromycin 500 mg, starting the day before surgery. All patients rinsed with chlorhexidine (0.2%) for 1 min immediately before surgery.

### 2.2. Intraoperative Care

All surgical procedures were performed under local anesthesia (2% mepivicaine with epinephrine 1:100,000 and/or 4% articaine with epinephrine 1:100,000).

The same surgeon, who had more than 25 years of experience in oral surgery, performed all surgical procedures.

All the teeth needed a surgical approach with a flap and bone surgery. To standardize the technique as much as possible, a triangularly shaped intrasulcular mucoperiosteal flap with lateral releasing incision in all cases was used. Bone removal was performed with a water-cooled bur in a high-speed surgical drill. Tooth sectioning was performed with a fissure bur in a high-speed surgical drill to reduce the entity of bone surgery.

In some cases of completely formed third molars with roots in a close relationship with the mandibular canal, piezo surgery was employed for bone surgery to reduce the risk of neurological damage. Rinsing of the alveolus with physiologic solution was performed before suturing with 4-0 silk. Primary closure of the flap was attempted in all cases of completely impacted third molars, while in cases of partially impacted third molars, we obtained a secondary intention healing. No dressings or hemostatic agents were used. The duration of surgery from the initial cut to the final suture was noted in the patient’s record.

### 2.3. Postoperative Care

Antibiotic therapy was continued for four days in the postoperative period (1 g or 0.5 g, depending on the patient’s weight, of amoxicillin–clavulanic acid every 8 h) [13]. In patients allergic to penicillin, we prescribed azithromycin 500 mg (one tablet a day for three days). Chlorohexidine solution 0.20% two times a day was employed in the two postoperative weeks.

Each patient was asked to fill out a VAS scale daily from the surgery to the suture removal visit and to report any discomfort [3].

Sutures were removed seven days after surgery and all patients were recalled at four weeks and at three months post-extractions. The visit at four weeks was to verify the maintenance of good oral hygiene and the absence of a socket infection (purulent discharge), while the visit at three months was made to verify the periodontal healing.

All patients had the opportunity to contact the dental office for any problem related to extraction, and in such cases, they were seen within a few hours.

Age, gender, panoramic radiographs, and Pell and Gregory classification were recorded on an Excel spreadsheet by a researcher independent from the surgeon; this researcher also reported the occurrence of delayed infections.

Delayed infection, as reported in a previous study [22], was “diagnosed in the presence of swelling and/or purulent drainage from the alveolus arising 15 days to 2 months post extraction”. In the early stages of infection, swelling (intraoral and/or extraoral) could be present without the purulent drainage. Infection was sometimes associated with fever. Pain, when present, was “moderate” [22].

### 2.4. Statistical Analysis

A chi-square test was performed aiming to compare patients and teeth with alveolitis between the incompletely formed third molar and completely formed third molar groups; the odds ratio and 95% confidence interval [25] were also computed using the Armitage and Berry method. The α level was a priori set at 0.05.

## 3. Results

Regarding immediate postoperative infection, two of the subjects presented an infection with drainage of purulent exudates from the alveolus at suture removal after 1 week, and one patient reported fever in the postoperative week.

We reported one case of transitory alteration of the sensitivity of the inferior alveolar nerve, completely resolved in three months, in a female (31 years old) after the removal of a partially impacted third molar.

Eight patients reported high pain in the first postoperative week. Four were third molars completely impacted and four were partially impacted.

We reported one case of manifest swelling that limited chewing and speaking.

Nineteen of the 265 patients (7.2%) presented a delayed infection between 2 and 8 weeks after extraction of the third molar with purulent exudate from the alveolus and swelling (Table 1).

Of these 19 cases of late infection, 18 developed in patients aged under 20 years (134 subjects), and they were all germectomies. The patient’s mean age was 15 years (range 12–20). The large age range was due to individual variation in tooth development.

We reported one case only of late infection in a patient older than 20 years with completely formed roots. The group with completely formed third molars (131 patients) had a mean age of 29 years (range 17–55).

In the group with completely formed third molars, one of the 131 patients presented a delayed infection after 22 days from the third molar extraction.

Table 2 quantifies the risk of developing alveolitis based on the Pell and Gregory class. Considering the patients, because alveolitis involved one tooth per patient (except two patients who had the complication after both germectomies), we can infer that patients with Pell and Gregory class III and II have a 20 times higher risk than class I to incur alveolitis (odds ratio = 20.17, 95% confidence interval 2.65–153.47)

Considering the tooth, the significantly higher number of alveolitis in patients who had germectomies (*p* = 0.0006) was confirmed by comparing teeth with alveolitis in the two groups; teeth with Pell and Gregory class III and II had a 16-times greater risk of incurring alveolitis than class I (odds ratio = 16.26, 95% confidence interval 2.16–122.48).

## 4. Discussion

Late infection can represent a true problem for a patient, because the late onset of pain and swelling and the need for a new antibiotic therapy could change his/her life in a period usually considered to be consolidating healing. In some cases, this complication may represent a medicolegal issue. Late infection has been investigated by a few authors that reported very different incidences [18,19,20,21]. In the study by Osborn et al. [19], the delayed infection rate was 3.7% in the overall patient sample (12–83 years), but the rate was 6.7% in the younger age group (12–24 years). Piecuch et al. [20] found a higher incidence of late infection in the case of complete bony impaction (3.7%). Figuereido et al. [21] found an increased likelihood of delayed infection (0.5–1.8%) following the extraction of mandibular third molars with total soft tissue retention and a lack of space distal to the second molar. Moreover, a vertical or mesioangular inclination, tooth sectioning, bone retention, and the depth of inclusion were also considered risk factors for this kind of infection.

In this study, we analyzed the teeth with the Pell and Gregory classification, commonly employed by clinicians when they evaluate completely formed third molars, while in a previous study [22], the evaluation was performed using the Ganns protocol [26]. Monaco et al. [22] hypothesized that a lack of distal space—Ganns ratio ≤ 0.5 (that corresponds to a Pell and Gregory class III)—would be the most important factor in the onset of delayed infection after the extraction of impacted mandibular third molars. The incidence of delayed infection in this study was higher (9.2%) than the incidences reported previously [21]. However, Figuereido et al. [21] suggested that lower third molars with total soft tissue retention and a lack of distal space were more likely to develop delayed infections. Moreover, they noticed that 70% of the delayed infections were related to asymptomatic third molars.

The high late infection rate found in the Monaco et al. study [22] was related to the fact that they explored all the critical factors (complete impaction, lack of distal space, and asymptomatic teeth) in a selected cohort of patients with a small age range (12–20 years). The sample size included in this study suggested the need to perform further studies with larger populations to better explore the relevance of the anatomic condition in the incidence of late infection.

We reported 19 cases of delayed infection in a sample of 265 patients (age range 12–55) with 7.2% of incidence. This incidence was lower than the previous Monaco study (9.2%) and similar to the incidence found by Osborn et al. (6.7%) [19]. Analyzing the results, this decrease appears to be related to the increased age of the patients, because we observed 9.2% of delayed infection in patients under 20 years of age and 0.8% of delayed infection in the older patients (20–55 years of age) with completely formed third molars.

Analyzing the patients, we observed 13% of delayed infection in subjects who underwent germectomies and 0.8% in subjects with completely formed third molars. This increasing percentage was related to the fact that two patients in the younger group, with extremely reduced space distal to the second molar, developed a delayed infection in both sides after third molar removal.

The more difficult surgical extractions performed in older patients (25 cases had a close relationship with the mandibular canal) were not associated with an increased risk of delayed infection as hypothesized by Figureido who described that osteotomy, tooth position, and need of sectioning appeared to be associated with an increased incidence of late infection [21]. Instead, the more simple surgical removal of the germs was associated with an increased incidence of late infection. Usually, germectomy is a less invasive surgery when compared to extraction of a completely formed tooth [1,2]. The germs were always sectioned into four fragments along two perpendicular lines, and so the amount of bone removal was very similar in all cases. In addition, complete impaction in the ramus, in the case of germ removal, did not require more osteotomy because a small cortical window of the same size (about 3 × 6 mm) was performed distobuccally to the second molar.

The young patients of this study who presented delayed infection, if analyzed using the Ganss protocol, showed a ratio between the distal space and the crown width ˂ 0.5 in 16 of 20 cases. This ratio corresponds to a class III of Pell and Gregory.

In the patients with completely formed third molars, the only case of delayed infection developed in a patient with a horizontally impacted third molar. In this case, the Ganss ratio was 0.78 (one of the more reduced spaces in this sample), and the tooth was completely impacted.

In the younger age group, we observed a more reduced space distal to the second molar, as documented by class III of Pell and Gregory. This anatomic condition could determine a low possibility of the patient maintaining good hygiene distal to the second molar, thus leading to a greater risk of food impaction. Moreover, the complete bone and tissue impaction of the germ make easier a first intention closure by the surgeon. However, first intention healing could determine, as stated by Waite and Cherala [27], the “one-way valve effect that allows food debris to enter the socket but not easily escape”, and this condition results in a greater likelihood of socket infection. In the present study, first intention healing was obtained in all cases in the germ group in the first week postoperatively, except in one case in which wound dehiscence occurred.

In the older age group, we had 102 cases of partially impacted third molars, and in these cases, we did not obtain a first intention closure. We could hypothesize that a second intention healing together with the presence of more space distal to the second molar, enable a “self-cleansing” of the wound, as stated by Waite and Cherala [27], and enable the patient to maintain good hygiene distal to the second molar, thus leading to a minor risk of food impaction.

We opted for primary closure of the surgical wound in the germ group because our attempt was to achieve complete control of postoperative bleeding in young patients. However, considering the results of the older group, where in most cases we obtained a second intention healing, it might be more appropriate to choose a suture-less technique to obtain a ‘self-cleansing’ socket when the distal space is extremely reduced.

Based on our findings, all patients should be informed of the possibility of this late postoperative complication. Furthermore, to reduce the risk of late infection, the surgeon, in cases of bilateral tooth impaction, should remove the second germ after a prolonged recovery period (we suggest 45–60 days). When the space is severely reduced, self-cleansing of residual food is particularly difficult, and the recommendation not to chew solid food for at least four weeks on the operated side could be useful to reduce the incidence of late infection.

## 5. Conclusions

This study confirmed that the space distal to the second molar was significantly correlated with delayed onset infection, defined as an infection occurring between 2 and 8 weeks postoperatively.

For the surgeon, it is important to be aware that the reduced anatomical space distal to the second molar could lead to a greater possibility of late infection. Furthermore, it is important to provide proper information to the patient, and—in case of third molar removal in both sides—a correct timing of the surgical procedures is critical.

## Figures and Tables

**Figure 1 ijerph-19-04028-f001:**
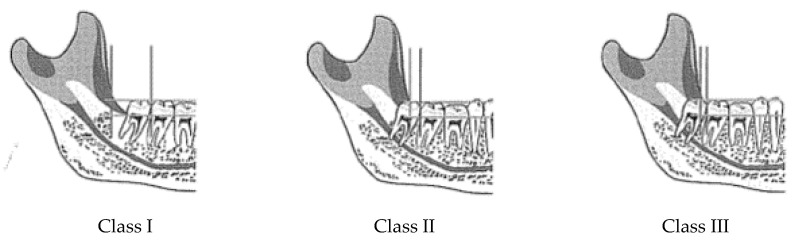
Pell and Gregory classification regarding the space distal to the second molar in relation to the difficulty of the third molar extraction. (**Class I**): the space is sufficient for the third molar eruption and the extraction is easy. (**Class II**): the space is not enough for a complete eruption and the extraction is more difficult. (**Class III**): the third molar is completely impacted, and the extraction is the most difficult.

**Table 1 ijerph-19-04028-t001:** Characteristics of the patients with delayed onset infection.

Patient Number	Case of Delayed Onset Infection	Gender	Pell and Gregory Class	Age (Years)	Length of Surgery (min)	Onset of Infection (Week)
1	#1	M	3	15	28	4
2	#2	F	3	14	29	4
3	#3	M	3	14	30	4
4	#4	F	3	15	28	4
5	#5	M	3	14	29	4
6	#6	M	3	16	27	2
7	#7	F	3	15	25	4
8	#8	F	3	16	30	4
9	#9	F	3	17	28	4
10	#10	M	3	14	25	4
11	#11	M	3	15	30	4
	#12		3		35	4
12	#13	F	3	15	28	4
13	#14	F	2	18	25	4
14	#15	F	2	17	24	3
15	#16	F	2	16	26	7
16	#17	F	3	15	29	8
	#18		3		30	8
17	#19	F	3	15	23	8
18	#20	M	2	15	45	8
19	#21	M	2	28	40	4

**Table 2 ijerph-19-04028-t002:** Risk of developing alveolitis when a patient and tooth belong to Pell and Gregory class II and III. Reference category: class I.

	Patient as Unit of Analysis	Tooth as Unit of Analysis
Odds ratio	20.17	16.26
95% Confidence Interval	2.65–153.47	2.16–122.48

## Data Availability

Data can be requested to the corresponding author.

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
