# Peer review of "Incidence of Delayed Infections after Lower Third Molar Extraction"

_ijerph, 2022, doi:10.3390/ijerph19074028_

Round 1
Reviewer 1 Report
- The abstract is unstructured and is required for a clinical study!
- The introduction is absolutely insufficient in terms of volume (14 lines) and quality. It is not possible to cite 17 authors in half a sentence and 5 more in the other half !!! It can not!!!
"Third molar extraction can determine immediate (first postoperative week) [1-17] or 24 delayed (from 2 to 8 weeks postoperatively) infections [18-22]."
- When quoting an author, the numbering is placed at the end of the citation in brackets, not in the middle of nothing.
“The study of Monaco et al. of 2017 [22] showed, in a sample of young patients, as this 35 complication was related to a reduced space distally to the second molar [23] and as the 36 percentage of incidence was higher to what reported from previous researchers [19- 21]. "
“The extractions were performed for orthodontic reasons or for the presence of at least one of the criteria defined from the N.I.H. [24] that represent the rationale for third molar removal (pericoronitis, periodontal disease, caries, root resorption, pericoronal cyst). "
N.I.H. - For each abbreviation there must be the full name, and in this case it should be mentioned what is meant by NIH consensus!
- In some places square brackets are used, in others ordinary ones! Look the authors guidelines!
"Delayed infection (2-8 weeks from surgery) 29 was more difficult linkable to the surgical procedure and probably for this reason has 30 been the object of minor attention from researchers (18-22)."
- All authors from the reference part must be mentioned in the introduction!
- Citing authors is not done in results, but in the discussion!
- Table 1 is material for statistical analysis, no result!
- I believe that the clinical trial is quite large in terms of number of cases and time, but it is presented in an unacceptable way. The article should be rewritten entirely according to the basic principles for scientific article, and to be formatted (because here the fonts, brackets, etc. diverge - the references are total chaos)!
- The conclusion should be separated part!
- There are some self-citation but they present that this study is related with previous one!
Author Response
1) The abstract has been structured.
2)
- The introduction was intentionally kept short because the literature about immediate postoperative complications was well known and already discussed. Instead the literature about delayed postoperative complication is limited to the cited four studies.
- As suggested, the numbers of the references have been placed with square brackets
- I.H was reported as National Institute of Health-U.S.A. first and then as N.I.H.
- Authors not mentioned in the text have been removed from the references
- We agree that in Table 1 there is a description of the patients but only those who had a complication after the surgery and that are results.
- As suggested, conclusions have been separated from the discussion
Reviewer 2 Report
I would like to thank the authors for their work. The topic of postoperative complications after third molar surgery is a critical issue for both clinicians and patients and this research will help them predict the increased risk of postoperative infection. The paper is well written, however the introduction can be improved with more elaboration. Maybe an explanation of Pell and Gregory classification and even adding a diagram to visualise it will help the broader readers of the journal to better understand the paper.
Regarding the methods, few information about the systemic risk factors of the patients, rather than their age is provided. For example it is not clear if smoking factor was also considered for the cases. Also the effect of surgical technique in the development of postoperative delayed infection should be discussed since third molar cases with different Pell and Gregory classifications have different surgical difficulty.
A suggestive hypothesis of the reason behind increased rate of postoperative infection in class 3 Pell and Gregory cases will enrich the discussion of the paper.
Author Response
- As suggested, a figure illustrating Pell & Gregory classification has been added.
- We added informations about smoking habits of the patients.
Reviewer 3 Report
The manuscript submitted by Giuseppe Monaco et al, investigated the correlation of the space distal to the second molar with the delayed onset infection after extraction of the impacted third molar.
The manuscript itself is well constructed.
The topic is good and adequate for this journal.
The introduction is short and clear.
In my viewpoint, I do not have any comments regarding the methodology and significance of this important issue covered in this work. However, it is necessary to expose more detail about the results.
The discussion is well organised, supported by the articles from literature.
The conclusions are correctly, but they should be written separately by the discussions.
There are a few spelling errors which need to be resolved.
The references are not according to the style of journal. Please check it.
Author Response
- As suggested, conclusions have been separated from the discussion
- Spelling errors have been removed since the text was revised by the Editorial service
- All the reference section has been modified following the editorial style
Reviewer 4 Report
This is a useful paper for clinicians. There are some points that are difficult to understand. The results that made a significant difference should be summarized in a table.
Considering the patients, because alveolitis involved one tooth for patient (except 139
two patients who had the complication after both the germectomies ) we can infer that 140
patients with Pell & Gregory class III and II risk 20 times more than class I to incur in 141
alveolitis (Odds Ratio =20.17, 95% Confidence interval 2.65-153.47) 142
Considering the tooth, the significative higher number of alveolitis in patients who 143
undertaken germectomies (p=0.0006) is confirmed by comparing teeth with alveolitis in 144
the two groups ; teeth with Pell & Gregory class III and II risk 16 times more than class I 145
to incur in alveolitis (Odds Ratio =16.26, 95% Confidence interval 2.16-122.48).
Author Response
Results were summarized in Table 2.
Table 2 Risk of developping alveolitis when patient and tooth belong to Pell&Gregory class II and II Reference category: class I

Round 2
Reviewer 1 Report
The reviced version is appropriate!
Reviewer 4 Report
This study has been modified appropriately. I think it is a valuable study.